# Fall Risk Assessment Using Wearable Sensors: A Narrative Review

**DOI:** 10.3390/s22030984

**Published:** 2022-01-27

**Authors:** Rafael N. Ferreira, Nuno Ferrete Ribeiro, Cristina P. Santos

**Affiliations:** 1Center for MicroElectroMechanical Systems (CMEMS), University of Minho, 4800-058 Guimaraes, Portugal; rafaelrjnf22@gmail.com (R.N.F.); nuno.fribeiro@dei.uminho.pt (N.F.R.); 2LABBELS—Associate Laboratory, 4710-057 Braga, Portugal; 3LABBELS—Associate Laboratory, 4800-058 Guimaraes, Portugal

**Keywords:** fall risk assessment, fall prediction, wearable sensors

## Abstract

Recently, fall risk assessment has been a main focus in fall-related research. Wearable sensors have been used to increase the objectivity of this assessment, building on the traditional use of oversimplified questionnaires. However, it is necessary to define standard procedures that will us enable to acknowledge the multifactorial causes behind fall events while tackling the heterogeneity of the currently developed systems. Thus, it is necessary to identify the different specifications and demands of each fall risk assessment method. Hence, this manuscript provides a narrative review on the fall risk assessment methods performed in the scientific literature using wearable sensors. For each identified method, a comprehensive analysis has been carried out in order to find trends regarding the most used sensors and its characteristics, activities performed in the experimental protocol, and algorithms used to classify the fall risk. We also verified how studies performed the validation process of the developed fall risk assessment systems. The identification of trends for each fall risk assessment method would help researchers in the design of standard innovative solutions and enhance the reliability of this assessment towards a homogeneous benchmark solution.

## 1. Introduction

Falls consistently rank as the second main cause of unintentional injury deaths worldwide [1]. About 684,000 fatal falls and an estimated 37.3 million non-fatal falls, which require medical attention, occur each year. The elderly aged 60 and over have the highest fall risk due to their increasingly reduced cognitive, physical, and sensory status [1]. Therefore, there is a major need to develop tools that enable the assessment of the fall risk of targeted aged populations in order to suggest evidence-based treatment interventions towards a safer gait and, consequently, lower the fall risk.

In the last years, fall-related research has increased its focus towards fall prediction relative to fall detection. While fall detection systems aim at alerting the subject and healthcare professionals whenever a fall takes place, fall prediction systems warn subjects before the fall event occurs [2]. Therefore, as fall prediction helps preventing the fall occurrence, it further reduces the harmful consequences of a fall. Furthermore, fall risk assessment systems, which are built to predict future falls, pave the way for an efficient fall prediction [3]. In this regard, fall risk assessment methods using different wearable sensory systems have been developed in order to provide quantitative measures towards an objective assessment of the risk of fall [4,5,6,7]. Fall risk assessment based on wearable sensors can be performed from a long-term perspective, in which sensor data is used to predict subject’s long-term fall risk based on clinical scale scores [8,9], or from a short-term approach, where data collected is used to detect pre-fall/unbalanced situations in real-time and consequently identify fall risk events [4,5]. Sensor-based fall risk assessment tackles some issues related to more traditional approaches to assess the fall risk, which mainly consist of qualitative, subjective, and oversimplified clinical scales or questionnaires [7,8,9,10]. Nevertheless, this sensor-based assessment is normally performed in supervised conditions, in which the behaviour adopted by test subjects may not be representative of the one adopted in the everyday life, as the subjects might be performing their “best effort” during the experimental tasks [11]. As fall events occur generally in an unpredictable fashion during the everyday life context, there is the need to assess the fall risk in uncontrolled conditions. An unsupervised fall risk assessment using wearable sensors would provide continuous monitoring during daily functional tasks and thus reflect subject’s real fall risk. Thus, the interest on wearable sensors has been increasing considering the monitoring of the fall risk among the elderly community. These wearable devices benefit from their wide range of products in the market, small size, as well as the meaningful data they provide while being an alternative low cost solution [6,7].

As mentioned in some previous review articles [2,3], reliable fall prediction and prevention require a multifactorial analysis according to the large amount of different factors that can cause fall events. Therefore, in order to build a reliable fall risk assessment system, both intrinsic and extrinsic factors have to be considered in the analysis. As reported by Rajagopalan et al. [2], intrinsic fall factors include characteristics inherently related to the subject such as its mobility impairments, neurological disturbances, age, or fall history. On the other hand, extrinsic factors are generally related to the environment in which the subject is inserted and account for inappropriate footwear, clutter, slippery surfaces, or poor lighting. Therefore, the complex interaction between biological, environmental, demographic, and behavioural fall risk factors require framework solutions that enable the integration of both contextual data regarding the environment and subject’s behaviours as well as physiological health information [2]. Indeed, literature studies have been exploring various kinds of statistically different features between fallers and non-fallers, which may have the potential to be monitored in fall risk assessment systems. Beyond the most common use of metrics extracted from kinematic and kinetic sensors to assess the risk of falling [6], other approaches included biosignal analysis by means of cardiovascular [12], electromyography [5,13] or electroencephalography [13] data. Additionally, sleep quality metrics can be monitored within the scope of fall risk assessment [14].

Rajagopalan et al. [2] indicated that current fall prediction systems are generally tested in laboratory conditions, which do not reflect the real relationship between the fall risk factors. As such, laboratory testing of fall risk assessment systems does not address the real fall risk and may bias the ability to predict future falls in regard to daily life unsupervised testing conditions [11,15]. Concerning this validation process, Howcroft et al. [16,17] pointed out the concern between the use of retrospective (fall history) or prospective (future falls) fall data as the standard to evaluate the predictive accuracy of fall risk assessment systems. Shany et al. [18] stated that, although many studies did not consider prospective falls, there has been an observed effort in recent years to incorporate future fall occurrence in fall risk assessment. As fall risk assessment models are built to predict future falls, the use of prospective fall occurrence information during the follow-up period after the baseline assessment may be more appropriate to validate the performance of the models [17]. Furthermore, retrospective fall occurrence is associated with the imprecise recall of past fall events by the test subjects, which may hinder the retrospective assessment [19]. Moreover, in retrospective fall risk assessment, as subjects have experienced previous falls, they may walk in a cautious way during the assessment due to fear of falling, which would deviate them from their natural gait [16,17]. Fear of falling produces observable changes in the gait and muscular activity patterns, such as increased double-support time, reduced stride length, and increased levels and duration of muscle co-contraction, as subjects seek to adopt a more stable gait to avoid another fall [20,21,22]. This fear may also minimise the execution of daily life activities, which leads to physical inactivity and consequent social isolation. These factors significantly correlate to the decrease in muscle strength, physical performance, and ability to control the posture [20]. As such, fear of falling leads to gait, balance, and cognitive disorders over time, resulting in mobility deterioration and consequently increasing fall risk [23]. Therefore, fear of falling emerges as an important psychological parameter in fall risk assessment, as the detection of the consequent motion deterioration could allow the identification of high fall risk subjects [20].

Recent reviews targeting fall risk assessment have presented and discussed the different approaches to analyse fall risk. For instance, Rucco et al. [6] reviewed the state of art of the fall risk assessment using wearable sensors investigating the most used sensor technologies, their number and location, as well as the number and type of tasks performed in the experimental protocol. Montesinos et al. [24] conducted a systematic review that studied the most significant and strong associations between combinations of feature categories, tasks performed and sensor locations to ascertain a subject fall status, as faller or non-faller. Rajagopalan et al. [2] performed a comprehensive review regarding the relationship between the different fall risk factors and highlighted current work and challenges on fall prediction systems. However, the analysis within these manuscripts was performed without specifying the different fall risk assessment methods, such as long-term or real-time fall risk assessment. Therefore, the identification of trends is less reliable than an individual analysis carried out for each fall risk assessment method identified. The assessment of the fall risk from both long-term and real-time perspectives requires different specifications and setups and, consequently, different and individual analysis. For instance, a specific type of sensor placed on a certain position of the body can be widely used for a specific fall risk assessment method and not for another. Furthermore, none of the previously mentioned reviews ascertained the validation processes carried out to validate the fall risk assessment systems found in the literature.

Thus, the aim of this work was to find evidence on the following topics: (i) “Which are the main types of fall risk assessment methods using wearable sensors in literature studies?”; (ii) “What types, number, and location of wearable sensors were adopted in the literature studies?”; (iii) “Which tasks or clinical scales were performed during experimental protocols for data acquisition?”; (iv) “Which algorithms are used in the scientific literature for the classification of fall risk?”; and (v) “How was the validation of fall risk assessment systems performed using wearable sensors?”. The first, fourth, and fifth questions offer novel analysis regarding the reviews articles [2,6,24]. To the best of the authors knowledge, no previous study has addressed the first question. The third question offers a technological description of the sensors used in fall risk assessment systems. This allows the further comparison with previous review studies to ascertain if trends of sensor specifications are maintained or updated. The fourth question offers a review of the tasks or clinical scale protocols performed for data collection.

The remainder of this narrative review is organised as follows: Section 2 describes the search strategy employed. Section 3 highlights the characterisation of the different fall risk assessment methods identified in the scientific literature and the methods used to validate fall risk assessment systems. Section 4 provides a general discussion of the search outcomes and points towards the future directions on the fall risk assessment field. Lastly, Section 5 presents the conclusions obtained from this review.

## 2. Methods

An electronic systematic search was accomplished in IEEE, Scopus, Web of Science, and PubMed databases on the topic of fall risk assessment of towards the elderly population using wearable sensors. The search was completed in the aforementioned databases on 3 November 2020. On IEEE the keywords used were: (aged OR elderly OR geriatric OR old) AND fall risk AND wearable sensor. The terms (aged OR elderly OR geriatric OR old) AND (wearable sensor OR wearable device) AND fall risk AND (gait OR posture OR walking) were used in the other 3 databases. In order to provide an overview of the most recent and emerging trends of fall risk assessment using wearable sensors, the search was conducted considering all articles that were published after 2015. A total of 332 articles were found and 223 remained after removing duplicates. Further, a careful reading of the title and the abstract of those articles enabled the exclusion of articles that clearly did not perform fall risk assessment or were a review. Reviews were excluded from the search results as the purpose of the search strategy was to find studies which developed a fall risk assessment system. Following this procedure, 48 articles remained for full text reading. In order to screen the most important ones, eligibility criteria were applied to the selected papers. Articles were excluded if: (i) the system described in the study presented any kind of non-wearable device; (ii) a fall risk assessment method was not applied or described; (iii) there was a lack of information on either the sensor system or its placement on the body; and (iv) the study was a previous version of a more recent one, being both in the 48 selected articles group. Regarding the application of these criteria, 16 articles were selected for further analysis. In Figure 1, it is depicted the Preferred Reporting Items for Systematic Review and Meta-Analysis (PRISMA) flowchart regarding the previously described literature search.

## 3. Fall Risk Assessment Methods

As suggested in Figure 2, the 16 selected manuscripts were divided into groups according to the method used to assess fall risk.

A group of nine studies [8,9,25,26,27,30,31,33,34] assessed fall risk from a long-term perspective based on clinical established scales. This group comprised more than half of the manuscripts, i.e., 56%. In addition, 25% of the selected manuscripts [4,5,28,29] considered fall risk assessment from a short-term or real-time approach by developing a system and an algorithm able to identify pre-fall/unbalanced situations and consequently detect fall risk events. Lastly, three studies [12,13,32], i.e., 19%, which followed different approaches to assess fall risk, were identified and included in the “Other Methods” group.

### 3.1. Fall Risk Assessment Based on Clinical Scales

Vieira et al. [33] developed a gamified application for the elderly to independently measure the Berg Balance Scale (BBS) score at home by means of a custom-made sensor containing an accelerometer and a gyroscope. Shahzad et al. [26] estimated the BBS score from data acquired from a single accelerometer. Tang et al. [9] performed a study to obtain the BBS and MiniBEST test scores for each subject with a sensor apparatus composed by a SmartShoe, which comprised a pressure sensitive insole with three pressure sensors and an accelerometer, as well as an hip accelerometer. Yang et al. [31] conducted four environment-adapting TUGs in order to assess fall risk in a more comprehensive way than standard TUG by adapting gait in complex environments. During the trials, subjects wore a Smart Insole (SITUG) in each foot, with a sensing device composed by 16 pressure sensors array along with an Inertial Measurement Unit (IMU) including an accelerometer, gyroscope, and magnetometer. Saporito et al. [27] attempted to predict a remote TUG score based on data recorded from three days of free-living conditions by means of one accelerometer and one barometric sensor. Buisseret et al. [30] assessed subjects’ fall risk based on the TUG test score and data acquired from an accelerometer, a gyroscope and a magnetometer during the 6-minute walking test (6MWT). Dzhagaryan et al. [34] developed a wearable system, the Smart Button, capable of providing an automated mobility assessment of TUG and 30-second Chair Stand (30SCS) tests from data collected by an IMU with an accelerometer, a gyroscope and magnetometer sensors. In both studies conducted by Rivolta et al. [8,25], the Tinetti test score was predicted for each of the test subjects by means of data collected from a single accelerometer. Further details about the sensor systems used are provided in Figure 3.

#### 3.1.1. Sensor System Characteristics

Figure 3 summarises the sensor characteristics from the studies that performed fall risk assessment based on clinical scales.

All the studies used at least one accelerometer, which underlines the importance of the use of acceleration data to characterise the score results from clinical standard scales. The use of gyroscope sensors was highlighted in four articles [30,31,33,34]. This search revealed that accelerometers and gyroscopes were the most widely used sensors for this fall risk assessment method. The magnetometer sensor is also included in the sensing device of three studies [30,31,34] and is used along with both acceleremeter and gyroscope sensors. Beyond inertial sensors, pressure sensors were used in two studies [9,31]. Concerning the sensors’ sampling frequency, all the studies acquired data from sensors at 100 Hz or less except Tang et al. [9], which used 400 Hz, and Vieira et al. [33] that did not mention the frequency adopted. However, in the data processing stage, Tang et al. [9] downsampled data from 400 Hz to 25 Hz.

Most of the studies used a small number of three sensors or less. However, Tang et al. [9] and Yang et al. [31] used 9 and 38 sensors, respectively. In their setup, Yang et al. [31] used 32 pressure sensors and 2 IMU’s (with accelerometer, gyroscope, and magnetometer). Tang et al. [9] sensing apparatus consisted on six pressure sensors and three accelerometers. Within these manuscripts, almost all sensors were placed in the insole of the test subjects, thus the high amount of sensors did not compromise the wearability of the system. All the single sensor solutions that assessed fall risk through clinical-based scales used accelerometers [8,25,26]. The most widely used two-sensor combination for fall risk assessment is accelerometer and gyroscope, which is line with the search results of Rucco et al. [6]. In addition, four articles used the accelerometer and gyroscope combination [30,31,33,34], with Buisseret et al. [30] and Vieira et al. [33] using only data from those two sensing modalities.

Furthermore, five studies described the sensor placement on the chest [8,25,27,33,34], two on the waist/lower back [26,30], two on the feet [9,31] and one on the right hip [9]. Both studies that considered the feet to place the sensors used pressure sensors [9,31]. Additionally, eight studies [8,9,25,26,27,30,33,34] considered at least one upper body part to place the sensors, in which seven of them only considered upper body parts [8,25,26,27,30,33,34]. The chest and the lower back were the most used upper body locations. Therefore, the upper body contains the preferred locations to place the wearable sensors in fall risk assessment based on clinical scales.

#### 3.1.2. Clinical-Based Scales Adopted

The variety of clinical-based scales adopted in the literature towards fall risk assessment is shown by the 6 different scales included in the group of 9 studies. TUG was the most selected scale [27,30,31,34] and BBS was the second most adopted [9,26,33]. The Tinetti test was implemented in both studies conducted by Rivolta et al. [8,25] and MiniBEST, 6MWT, and 30SCS were included in one study each [9,30,34]. In addition, three studies conducted two different clinical scales [9,30,34]. While the majority of the studies [8,9,25,30,31,33,34] collected data from activities performed during the clinical scales experimental protocols to assess fall risk, some collected data from activities outside the clinical scale protocols. For instance, Shahzad et al. [26] attempted to predict BBS score of test subjects by means of data collected during a routine which included a group of simple physical movement activities, namely the TUG test, five times sit-to-stand test, and alternate step test. Further, in Saporito et al. [27] data collected from subjects during 3 days of free-living conditions was used to predicted TUG time score.

#### 3.1.3. Algorithms for the Classification of Fall Risk

In this fall risk assessment method, four studies implemented Machine Learning models [8,9,26,27], two considered a Deep Learning approach [25,30], two adopted threshold-based algorithms [30,33], and two studies did not perform this classification [31,34].

All four studies which applied Machine Learning used linear regression-based models to predict clinical scale scores. Shahzad et al. [26] used linear regression Machine Learning models to estimate the scores of the BBS test from the information provided by a single accelerometer positioned in the lower-back. In the same study, researchers opted to choose Machine Learning models that could be applied in small datasets and found that linear least square and LASSO regularised linear regression outperformed decision tree-based models, especially the LASSO one. Saporito et al. [27] also adopted a regularised linear model for the estimation of a TUG score, by means of signals collected from an accelerometer and a barometer in free living conditions for 3 days. Moreover, Rivolta et al. [8] applied a multiple linear regression model in order to predict the value of the Tinetti test scores assigned to the subjects by a clinician, using data obtained from a single sternum-mounted accelerometer. Tang et al. [9] applied a linear kernel support vector regression to predict clinical scores of BBS and MiniBEST from pressure and acceleration sensors data.

Some authors considered the use of Deep Learning [25,30]. Rivolta et al. [25] attempted to estimate the Tinetti test scores based on gait and balance features obtained from a single low cost acceleration sensor, considering a two-fold problem: (i) a binary classification problem to dichotomize individuals at score 18 as High and Low Fall risk; and (ii) a regression problem in order to estimate the gold standard Tinetti score assigned to each subject. Based on the performance results, the Artificial Neural Networks (ANN) provided better classification outcomes than the linear model.

Buisseret et al. [30] implemented a Deep Learning model, as well as a threshold-based algorithm in order to predict the risk of falls based on the TUG and 6MWT. Therefore, a 6-month prediction of subjects’ fall risk based on prospective fall occurrence as the start of the study was performed in three different classification ways: (i) a threshold-based approach considering only the time taken to complete standard TUG; (ii) another threshold-based approach (TUG+) considering the previously described time and kinematic parameters computed from IMU sensor data; and (iii) a Deep Learning Convolutional Neural Network (CNN) network that receives the raw IMU data only. The authors verified that both TUG+ and the Artificial Intelligence (AI) algorithm enhanced the performance in several classification metrics of the faller status of the subjects regarding the standard TUG alone. Vieira et al. [33] also implemented a threshold-based approach in order to assess the score of BBS through accelerometer and gyroscope measures. The researchers established reference values concerning each of the movements performed during the test in order to assign their respective classification. The works developed in [31,34] assessed the performance metrics of the features calculated by their systems against ground truth measures of video and optical motion capture system, respectively, rather than using algorithms to classify subject’s fall risk.

### 3.2. Fall Risk Assessment Based on the Detection of Fall Risk Events

Besides the clinical scale-based approach, four manuscripts [4,5,28,29] addressed fall risk assessment from a real-time perspective, focusing on the detection of fall risk events during the performance of activities. The details about the sensor systems used are presented in Table 1. Saadeh et al. [4] used the data collected from an acceleration sensor to distinguish between ADLs and pre-fall events. Their system achieved a timely prediction of fall events, activating a fall risk alarm before the fall occurrence. Rescio et al. [28] described an EMG-based system composed by four EMG sensors capable of detecting and recognising fall risk events. Leone et al. [29] also presented a four EMG sensor-based fall risk assessment system capable of recognising pre-fall events. Later, the authors developed a smart sock system, each one equipped with two EMG sensors, able to detect unbalance events associated with a potential fall risk [5]. More details about the performance metrics obtained by these systems are further provided in Table 2.

One important aspect analysed by each of the four studies was the lead-time. This time, which was used to study system’s detection performance of fall risk events, was considered with two different meanings. Saadeh et al.’s investigation [4], as well as both studies conducted by Leone et al. [5,29], evaluated detection performance of the system considering the lead-time as the time between the detection of the unbalance event and the impact of the fall. Saadeh et al. [4] mentioned that their system could predict a fall event with a lead-time between 300 ms and 700 ms before the fall impact. Leone et al. [29] claimed a mean lead-time of 775 ms of their system and, in a later study performed by the same authors [5], a smart sock EMG system was able to detect unbalance conditions with 750 ms of mean lead-time. However, Rescio et al. [28] interpreted lead-time from a different perspective, by considering it to be the time delay between the onset of the perturbation and the instant when the perturbation was detected. The authors claimed that their system was able to detect a perturbation 200 ms, on average, after its onset.

#### 3.2.1. Sensor System Characteristics

Table 1 depicts the sensor characteristics adopted in the studies that performed fall risk assessment based on the detection of fall risk events.

EMG-based systems were used in three studies [5,28,29] to detect pre-fall scenarios or unstable situations associated with fall risk. On the other hand, Saadeh et al. [4] described the detection of fall risk events based on accelerometer data. All the studies collected data using sampling frequencies higher than 100 Hz. All sensor systems were composed of four wearable sensors or less. A single-sensor solution comprised by one accelerometer was used in [4], two EMG sensors were used for each smart sock in [5], and a system with four EMG sensors was presented both in [28,29]. Saadeh et al. [4] placed the accelerometer sensor in the upper thigh. The three other studies placed EMG sensors in the *gastronecmius* and *tibilias* muscle groups. Leone et al. [5,29] specified the use of these sensors in the *gastrocnemius lateralis* and *tibialis anterior* muscles.

#### 3.2.2. Types of Activities Performed

In order to collect data to identify fall risk events, the four studies performed ADL and fall events in the experimental protocol. Rescio et al. [28] instructed test subjects to simulate a series of events in a random order: (i) being at idle position or walking, both in either a normal context or presented with a deviant auditory stimuli; (ii) perform some common ADLs such as bending, lying down, standing up or sitting down; and (iii) unstable situations provoked by a tilting platform which simulated loss balance characteristic of fall events. Saadeh et al. [4] adopted an experimental protocol similar to the one performed to obtain the MobiFall dataset [35] and used the collected data along with the data from MobiFall dataset to train and test their system. A total of six different examples of falls and 11 ADL events were performed. ADLs included events that have a higher chance of being classified as false positives/falls such as: (i) jumping and jogging, as they are abrupt events that are alike to a fall event; (ii) stepping in a car or sitting on a seat; and (iii) performing standing or walking tasks and ascending or descending stairs. In addition, forward lying falls, back chair falls, front knees falls, and side falls were considered in the protocol. In [29], Leone and colleagues also developed a dataset consisting of ADLs and fall events to train and test their algorithm. Although the types of ADL performed were not specified in the study, the researchers mentioned that the falls were provoked through a movable platform to cause unstable events in the test subjects. In a later work performed by the same authors [5], simulated ADLs and fall events were conducted in order to acquire data to train and test their algorithm. Simulated ADLs included: (i) walking; (ii) sitting down on a chair; (iii) bending; and (iv) lying down on a mat. Additionally, forward, lateral, and backward falls were induced by the same movable platform described in [29].

#### 3.2.3. Algorithms for the Classification of Fall Risk

Within the four studies that assessed fall risk from a real-time perspective based on the detection of fall risk events, three adopted Machine Learning models [4,5,29], whereas the remaining study used a threshold-based model [28].

Saadeh et al. [4] implemented a prototype system with two parallel real-time operating modes: slow mode fall detection (SMFD) and fast mode fall prediction (FMFP). In the FMFP mode, a nonlinear support vector machine classifier is used in order to predict fall events. This prediction is Patient Specific (PS) as, in the offline training stage of the classifier, PS parameters are computed and then uploaded to the system’s repository. Once those parameters are uploaded, they are used in the classification phase of fall prediction, adapting this process for each subject. Leone et al. [29] also implemented Machine Learning in order to distinguish between pre-fall and non pre-fall events. A linear discriminant analysis classifier was used to achieve a high generalisation capacity in the classification process while requiring low computational costs. Furthermore, in [5], Leone et al. used the same classifier to detect fall risk events using data collected from their developed smart EMG sock system. Rescio et al. [28] assessed the fall risk through a threshold-based approach as they had chosen the assurance of the system’s real-time operation rather than its generalisation ability.

### 3.3. Other Fall Risk Assessment Methods

There were other approaches also identified to assess the risk of fall. Selvaraj et al. [32] highlighted the importance of analysing the foot clearance during stair negotiation, as reduced values of this metric have an explicit mechanism linked to falls by increasing the chance of tripping. Therefore, the authors developed a wearable system for the subject’s shoe to determine the foot clearance during stair negotiation. The system was equipped with two distance sensors and an IMU sensor composed by an accelerometer, a gyroscope, and a magnetometer. Annese et al. [13] underlined the complexity of fall risk assessment and the need to perform it in a multifactorial approach in an everyday life monitoring scenario in order to accurately predict future falls. Hence, the same authors developed a cyber-physical system composed by EMG and EEG sensors interfaced to a Field-Programmable Gate Array (FPGA) responsible to perform an online processing of a subject’s fall risk coefficient. This fall risk index is based on a multifactorial approach considering the partial sum of four indexes namely, a subject condition or baseline factor, an environmental factor, an EMG co-contraction factor, and an EEG signal factor. While the first two factors, which are PS, are constant, the latter two are re-calculated just after a new step is detected during gait. Parvaneh et al. [12] explored the relationship between fall risk and the number of Premature Ventricular Contractions (PVC) episodes per hour, by using an ECG sensor.

### 3.4. System’s Validation

From the 16 selected studies, only 11 performed the validation of their fall risk assessment system [4,5,8,9,25,26,27,28,29,30,31]. As depicted in Table 2, the validation carried out on the fall risk assessment systems varied across these different studies. The fall risk outcome of the system was compared against reference measures in order to compute the system’s performance metrics.

Seven studies [4,8,9,25,26,27,30] validated their fall risk assessment systems using data collected from elderly patients, while the remaining four manuscripts used data from young subjects [5,28,29,31]. In addition, the number of subjects enrolled in the experimental protocols was usually equal or below 30 subjects [5,8,9,26,28,29,31]. Only four studies [4,25,27,30] included data from more than 30 subjects in their validation process. Saadeh et al. [4] was the only study that performed an external validation, i.e., used data collected outside the study’s experimental protocol to validate the system. As well as the data collected from 20 subjects (aged between 65 and 70) within their study, these authors also used data from 57 subjects (aged between 20 and 47) from the MobiFall dataset [35]. The remaining studies performed only an internal validation, i.e., validate the system using only data collected within the same study.

Cross-Validation (CV) was the most used validation method using both K-fold [26,28,29] and Leave-one-out [8,9,27]. The Holdout validation method was used in three studies [5,25,30]. Saadeh et al. [4] did not explicitly mention the validation method used. Lastly, Yang et al. [31] performed validation without using an algorithm. Their validation process consisted of comparing the features extracted from their smart insole system during the performance of four environment-adapting TUGs against video ground truth references.

Concerning the references measures for classification, five studies [8,9,25,26,27] used the clinical scale scores obtained at the baseline assessment as the reference measures for comparing the algorithm’s classification outcome. The algorithms developed by these 5 studies attempted to estimate the baseline clinical scale scores based on the wearable sensor data collected from the subjects. A group of four studies [4,5,28,29] labelled the data based on the activities performed. Thereby, data samples were labelled as fall risk/pre-fall or normal/ADL events and were used as the reference values to compare against the algorithm’s outcome. The algorithms developed in these studies attempted to detect if the subject was experiencing a fall risk event and obtain the lead-time values of that detection. Buisseret et al. [30] followed a different approach by considering the faller status, i.e., faller or non-faller, associated to each subject based on the prospective occurrence of falls during a follow-up period of 6 months. This faller status served as the reference metric for the algorithm’s fall risk outcome. Yang et al. [31] used video recordings to obtain reference values. The features extracted by their smart insole systems are compared against these reference values to obtain the system’s performance metrics. According to Table 2, the accuracy, sensitivity, and specificity were the most used performance metrics to validate fall risk assessment system’s performance. Nevertheless, the mean error is also used by some studies that predicted clinical scale scores [9,26,27]. Generally, studies seem to have reached good performance from the developed fall risk assessment systems.

## 4. Discussion and Future Directions

### 4.1. Which Are the Main Types of Fall Risk Assessment Methods Using Wearable Sensors in Literature Studies?

Concerning the search results, two main methods to assess the fall risk were identified. The first and most widely used consisted on the long-term assessment of fall risk and was based on clinical scales. In this method, which was adopted by nine studies, data from wearable sensors is used to predict subject’s fall risk based on clinical scale scores. Thereby, subjects are assigned to either high or low fall risk category. This method will promote the decrease in long-term fall risk by enabling subjects to continuously perform long-term fall risk assessments.

The second method, which was described in four studies, comprised a real-time assessment of fall risk by means of the detection of fall risk events. Data from wearable sensors was used to detect pre-fall/unbalanced situations in order to identify fall risk events. This method will promote the decrease in short-term fall risk by allowing subjects to be monitored in real-time on a daily basis, providing subjects feedback as to when a fall risk event is taking place. All the studies within this fall risk assessment method analysed the concept of lead-time. Two different perspectives of lead-time were considered: (i) the time between the detection of the unbalance event and the impact of the fall [4,5,29]; and (ii) the time delay between the onset of the perturbation and the instant when the perturbation was detected [28]. The first definition of lead-time may be particularly interesting, because if the time is high enough, it may enable the trigger of protection systems or alarms to reduce the harmful consequences of a fall [36]. In addition, the second concept of lead-time appears to be oriented to the speed of unbalance event detection rather than time for prevention of a fall. Future work in fall risk assessment should attempt to address both time concepts in order to evaluate not only the time for triggering a system for fall prevention, but also the speed of detection of unbalance events.

Another group of three articles, which assessed the risk of falling from other perspectives, was also identified [12,13,32]. Although these studies adopted interesting metrics and approaches to assess the risk of falling, they present some limitations. Selvaraj et al. [32] and Parvaneh et al. [12] only considered one metric to assess the fall risk and thus their studies did not perform a comprehensive fall risk assessment. Nevertheless, the inclusion of the foot clearance feature in fall risk assessment systems is pertinent, as it may depict the propensity of a subject to trip events [37]. In addition, cardiovascular metrics may also be important, as they can be considered a fall risk factor [3]. The cyber-physical system developed by Annese et al. [13] may bring some wearability issues, as users may not be comfortable with using EEG electrodes on a daily basis. In addition, considering that the baseline and environmental factors are constant, the assessment of fall risk based on these factors may not be accurate in all scenarios, as they are subject to change in real-life conditions.

Regarding the search results obtained, it was possible to conclude that the selection of which fall risk assessment method to adopt is strongly linked to the purpose of the assessment. For instance, if it is intended to perform a long-term prediction of the subject’s risk of falling, the estimation of clinical scale scores may be the most suitable approach, as it is performed in a single time period and allows direct feedback of fall risk based on the score obtained from the assessment. Further, it is possible to compare clinical scores obtained from the current and previous assessments in order to perceive the effectiveness of the evidence-based treatment interventions applied. On the other hand, if the objective of the assessment is a real-time prediction of the fall risk in the everyday life scenario, the method to detect fall risk events may become the most appropriate. Thereby, it is possible to monitor subjects continuously and alert them when fall risk events are identified.

### 4.2. What Types, Number, and Location of Wearable Sensors Were Adopted in the Literature Studies?

Inertial sensors, especially accelerometers, were used in all the studies that performed fall risk assessment based on clinical scales. As mentioned by Rucco et al. [6], the trend for using acceleration sensors may be related to the wide range of these inertial sensors on the market, as well as its low-cost and small size and weight. In addition, accelerometers have a lower power consumption compared to other inertial sensors, such as gyroscopes, which makes them more suitable for continuously monitoring applications [4,38]. In addition, as moderate correlations in scientific literature have been found between accelerometry features and some clinical scales, the use and interest of wearable sensors to assess the risk of falling through clinical-based scales has been growing [18,25]. Although three studies [8,25,26] only used accelerometers, four studies combined accelerometer with other inertial sensors, namely gyroscope [30,31,33,34] and magnetometer [30,31,34]. The stand-alone use of the described inertial sensors may bring various sources of measurement errors. For instance, in dynamic activities, accelerometers lack the proper estimation of orientation as they measure the motion’s external acceleration besides the gravitational acceleration. Additionally, due to gyroscope’s cumulative measurement errors, its use for estimating orientation in long-time activities may not be effective. In addition, especially in indoor environments, the geomagnetic field measures from the magnetometer are affected by ferrous structures [39]. Thus, the use of accelerometer, gyroscope, and magnetometer in a single IMU enables their sensing data fusion, which may solve the mentioned drawbacks and provide a reliable orientation estimation [39]. Furthermore, IMUs can be easily attached to subject’s clothing, which enhances the wearability of the sensor systems [30,31]. As such, IMUs became a reliable solution for gait analysis and, consequently, the assessment of fall risk. Pressure sensors were also included in two studies [9,31] to assess fall risk through clinical scales. Kinetic data collected from these sensors enable the detection of foot–ground contacts due to the pressure increase during specific phases of the gait cycle. This method of phase detection may be more accurate than the methodologies that use IMU sensor data, as contact phases are indirectly detected from inertial data by using foot orientation information [31,40]. Therefore, the use of data collected by pressure sensors in the feet insole may be helpful to enhance the performance of fall risk assessment. As opposed to fall risk assessment based on the detection of fall risk events, no study described the use of EMG sensors in fall risk assessment based on clinical scales.

There was also found to be clear evidence regarding the use of the wearable sensors on the upper body in fall risk assessment through clinical scales. Nevertheless, both studies that included pressure sensors in their systems placed these sensors on the feet [9,31]. According to Rucco et al. [6], the upper body placement of sensors is preferred over the lower limbs, as the upper body is preponderant in both static and dynamic stability, and is strongly linked to the upright gait which requires the ability to maintain upper body’s balance during walking. The chest and the lower back are the most adopted upper body locations to place the wearable sensors. Rivolta et al. [8] focused on the global body stability by placing their single wearable sensor on the chest, which restricts the relative motion between the body and the acceleration sensor. Shahzad et al. [26] and Buisseret et al. [30] considered the placement of sensors on the lower-back. In fact, the lower back positioning of wearable sensors is relevant in fall risk assessment applications as it is near the Center of Mass of the human body. Therefore, the sensors placed near that location provide signals with information of the whole body movements [26,41]. This evidence allows for wearable sensors to be included in user-friendly systems, e.g., waistbands, which can enhance the compliant use of the sensor systems by the elderly on a daily basis.

On the other hand, EMG sensors were the most used to detect fall risk events in real-time, being adopted in three of the four studies gathered [5,28,29]. The remaining study [4] used accelerometer data to perform this detection, activating a fall risk alarm whenever a fall event was predicted. As stated by Leone et al. [5], most of the studies in the scientific literature use inertial sensors to assess the fall risk. As such, the authors suggested the alternative to assess the unbalance condition by means of muscle contractile EMG data from the lower limb muscles. Concerning the search results, it seems that EMG signals may provide important information towards real-time fall risk assessment. In the three studies that used EMG systems to asses the fall risk [5,28,29], it was suggested that using lower limb surface electromyography sensors would promote higher lead-times than using inertial-based sensors, considering that the sudden change of EMG patterns due to an unbalance event is faster than the change of inertial signal patterns. However, the use of conventional EMG sensors may cause discomfort to the users on a daily basis, as they require a proper attachment to the surface of the skin next to the target muscle. This may bring compliance issues with the electrodes’ gel considering a long-term use of these kind of wearable devices. To overcome these drawbacks, Leone et al. [5] used hybrid polymer electrolytes-based electrodes, instead of the conventional pre-gelled electrodes, incorporated in socks to reduce skin irritation while improving biocompatibility, mechanical properties and signal detection. These novel solutions may increase users’ conformity with the use of EMG sensors and enhance its role in fall risk monitoring in free-living context.

Regarding sensor placement, it was observed that all the studies that used EMG sensors [5,28,29] considered its placement on *gastrocnemius* and *tibialis* muscle groups of both legs. These muscles are particularly important due to their role on walking, controlling stability, and maintaining the standing position. They are also relevant to evaluate gait changes related to age, fall risk, and postural deficits [5,29,42,43]. As *gastrocnemius* and *tibialis* are agonist–antagonist muscles, during a normal walk, they are alternatively activated. By detecting simultaneous and persistent activation of these muscles, it is possible to identify an unbalance event [22].

The sampling frequency adopted by each fall risk assessment method was different. While studies that assess fall risk based on clinical scales adopted frequencies below 100 Hz, the real-time detection of fall risk events was performed by acquiring data at a sampling frequency higher than 100 Hz. As the onset of fall risk events happen in fractions of a second, real-time fall risk assessment systems require sensor systems capable of collecting and processing high amounts of data in short periods of time. Therefore, a high sampling frequency is needed [4]. On the other hand, the analysis of long-term fall risk does not need to fulfil such requirements considering that the subject is not in danger of falling during the assessments. In addition, four studies [8,25,26,27] used sampling frequencies equal to or below 50 Hz. The use of lower sampling frequencies in this fall risk assessment method may be based on the fact that human activity frequencies lie between 0 and 20 Hz with 98% of its Fast Fourier Transform (FFT) amplitude contained under 10 Hz [44]. However, as lower sampling frequencies do not capture some useful particularities of the gait pattern, such as the subject’s walking style, higher frequencies may still be necessary to further enhance the reliability of metrics extracted for long-term fall risk assessment [45,46].

Regarding both fall risk assessment groups, there was found a clear evidence to use the least number of sensors, explained by the fact that most of the studies have developed systems with four wearable sensors or less. The technological advances in wearable sensors along with the meaningful data they provide are responsible for enhancing the wearable properties of fall risk assessment systems while maintaining or improving their performance.

Considering the search results, some important advantages are assigned towards the use of wearable sensors for fall risk assessment, as they: (i) increase the objectivity of the evaluation: (a) the assessment is based on objective data collected from sensors; (b) in conventional clinical scale assessments, participants are more aware that they are being evaluated and their behaviour may not be representative of the one in everyday context; and (c) it is removed the bias associated with the inter-operator variability of score assignment of conventional clinical scale assessments; (ii) enable the performance of some clinical standard scales at home, which increases the accessibility of these tests and decreases their related health care costs; and (iii) enable the real-time assessment of fall risk based on data collected during functional tasks performed in the everyday life context, which reflect subject’s real fall risk more accurately, and further allow for the timely detection of fall risk events.

Some of the findings in this search are in line with Rucco et al. [6], as: (i) the trend to use the upper body sensor placement, particularly of inertial sensors, was identified; (ii) the use of a single accelerometer was the more widespread single-sensor solution; and (iii) the combinations of the accelerometer sensor with either gyroscope or pressure sensors were the most used two-sensor solutions.

### 4.3. Which Tasks or Clinical Scales Were Performed during Experimental Protocols for Data Acquisition?

Considering the activities performed for data acquisition, the majority of studies [8,9,25,30,31,33,34] from the group of fall risk assessment based on clinical scales instructed their participants to perform experimental protocols relative to one or more clinical standard scales. The variety of clinical scales addressed in fall research is depicted by the six different scales adopted in the previously mentioned group of studies. According to the search results, the most adopted clinical scales were the TUG [27,30,31,34], the BBS [9,26,33] and the Tinetti test [8,25]. Although TUG is simple to administer in the older population, this test comprises some limitations, mainly due to its simplicity, which leads to the lack of information about gait adaptability that is strongly linked to fall risk [31,47]. This led Yang et al. [31] to conduct four environmental adapting TUG tests in order to obtain a more in-depth fall risk assessment. Other clinical scales, such as BBS and Tinetti, involve a more comprehensive group of activities, which may lead to a more representative amount of information on the subject’s fall risk [48,49]. Nonetheless, the time, material resources and monitoring from health care providers are more costly, making it less likely to be performed frequently and in the home environment. In order to overcome these issues, Vieira et al. [33] developed a gamified application that enables them to safely and autonomously perform the BBS. Nevertheless, as no results have been presented in the paper, there is no actual proof of the usability of the developed method. Despite being only considered in one study, the miniBEST test includes some advantages over the other clinical scales, as it evaluates more components of dynamic stability such as standing on a compliant or inclined surface and reacts to postural perturbations and crossing obstacles [9,50]. Concerning the 6MWT, as it provides relevant information concerning subject’s functional capacity, endurance, and systems involved during physical activity while requiring a simple setup, it may be interesting to include this test in fall risk assessment applications [51]. Although 30SCS requires a simple setup requirement, the test only provides the number of stands performed in 30 s as the only quantitative outcome [34,52]. It is noteworthy that three studies assessed the risk of fall using more than one clinical-based scale [9,30,34]. This can be particularly useful to gather metrics that are task-specific for each scale, which may enrich the information extracted to assess the fall risk. The decision of which clinical scale to adopt depends not only on the aim of the assessment, but also on the characteristics of the targeted population. Each scale has a specific objective and a preferable target population, both of which should be considered during the clinical scale selection. On the other hand, a minority of two studies [26,27] acquired data outside the clinical scale experimental protocol to predict the clinical scale score. This may be particularly useful if: (i) the activities used to collect data require less time than performing the clinical scale protocol [26]; or (ii) data acquired from free-living conditions could be used to predict a clinical scale score [27]. Therefore, more compliant ways to assess the fall risk can be achieved by decreasing the inconveniences associated with the performance of the whole clinical scale protocols. This should be addressed in future investigations.

The experimental protocol of studies that assessed fall risk based on the detection of fall risk events generally included some common ADLs, ADLs that can be misclassified as falls and fall events in different directions [4,5,28,29]. The inclusion of ADLs that can be misclassified is particularly interesting to test the algorithms’ fall positive rate and show its capability in classifying only true fall events. In [4], the conducted experimental protocol was similar to the one used to obtain the MobiFall dataset [35] and, along with the data collected in their study, they used data from that dataset in order to evaluate their system. The other studies from this fall risk assessment method [5,28,29] only included data collected within their experiments, which would limit the reliability of the systems’ performance metrics obtained. In addition, Leone et al. [5,29] and Rescio et al. [28] lack on the variety of ADL and fall events performed and on the number of subjects enrolled in the experimental protocol, in comparison to the study performed by Saadeh et al. [4]. Nevertheless, all the activities performed in these four studies were conducted in controlled conditions, which will introduce some bias on the data collected regarding real-world ADLs and falls. Future work should attempt to introduce real-world data from both ADL and fall event towards fall risk assessment based on the detection of fall risk events.

### 4.4. Which Algorithms Are Used in the Scientific Literature for the Classification of Fall Risk?

Concerning the analysis of the algorithms used for the classification of fall risk, Machine Learning models were the most used in the fall risk assessment methods identified [4,5,8,9,26,27,29]. These models are able to generate more reliable and reproducible results of fall risk classification than simpler algorithms such as threshold-based methods [3]. Aziz et al. [53] compared the performance of five Machine Learning algorithms against five threshold-based algorithms described in the literature to distinguish fall events and non-fall events. Accelerometer data from young adults was collected while performing eight types of ADLs, five types of near-falls, and seven types of falls in laboratory controlled conditions. The authors concluded that Machine Learning algorithms had generally greater performance than the threshold-based algorithms by providing higher values of sensitivity and specificity. The use of Machine Learning may be particularly useful in cases where it is complex to define a threshold value to classify data samples. However, if the threshold definition is simple and effective, threshold-based algorithms could be considered. As a matter of fact, Aziz et al. [53] found that two threshold-based algorithms had a lower false alarm rate than the Machine Learning algorithms. In this regard, the authors suggested that both algorithms could potentially be combined to increase the classification performance.

Nevertheless, Deep Learning algorithms have also been used to assess the fall risk and address some of the drawbacks related to the commonly used Machine Learning methods. Yu et al. [54] highlighted that the simple architecture of traditional Machine Learning models consists of only one layer that performs the extraction of a feature space from the raw input signals. However, the information processing mechanisms exhibited by humans indicate a more complex processing of the sensory input information, suggesting that data processing is performed through layered hierarchical structures [54]. Therefore, Deep Learning algorithms may be more appropriate to assess the fall risk, as they extract the most relevant features automatically towards this assessment. Hence, the manual extraction of pre-determined features from the sensor data, needed in traditional Machine Learning methods, is not required [55]. Deep learning models have been compared against traditional Machine Learning algorithms and have been shown to provide greater results, e.g., in gait event detection using accelerometer data [56]. The increasing computational power of micro devices over the recent years may lead to the implementation of more complex and sophisticated AI algorithms in wearable devices, which would enable an enhanced performance of fall risk assessment in a free-living context.

### 4.5. How Was the Validation of Fall Risk Assessment Systems Performed Using Wearable Sensors?

Different approaches were adopted to validate fall risk assessment systems, regarding the 11 studies that performed the validation process [4,5,8,9,25,26,27,28,29,30,31]. Most studies that performed fall risk assessment based on clinical scales used data from elderly subjects to validate their systems [4,8,9,25,26,27,30]. However, only one study that performed fall risk event detection used data from elderly participants [4]. Those remaining which used this method collected data from young subjects [5,28,29]. The participation of younger subjects may have been related to the compliance issues of elderly participants, considering the EMG sensor placement, compared to inertial sensors, do not require proper attachment to the skin. Nevertheless, future work on this fall risk assessment method should address the elderly’s muscle behaviour towards the detection of fall risk events, as the elderly are the targeted population for fall risk assessment. Regardless of the fall risk assessment method adopted, the number of subjects enrolled in the experiments was generally reduced. This will directly affect system’s performance metrics, as the reduced amount of subjects could not be representative of the whole population. Therefore, the algorithm’s classification can be biased to the study’s participants and not reproduce a reliable fall risk assessment towards subjects outside the study. Thus, researchers should focus on training and testing these algorithms on a larger sample of subjects.

The lack of external validation performed in the selected studies is remarkable. In fact, Saadeh et al.’s [4] was the only study which conducted an external validation, which was accomplished by using data from a public dataset, MobiFall [35]. Evaluating the performance of a system with data collected outside the study’s experiments would increase the reliability of the classification outcomes by reducing the bias of the system’s classification towards data collected within the study. This external validation should be pointed out as one of the main requirements in the design and conception of every fall risk assessment system [18]. The use of public datasets may be an interesting approach to perform external validation, particularly for fall risk assessment based on the detection of fall risk events. Choosing the datasets to perform the external validation must be done carefully and critically. Some recommendations should be followed during the dataset selection process, as pointed out by Casilari et al. [57]. By analysing some of the public available datasets, the authors suggested that the performance of a system should be evaluated by more than one dataset, giving the heterogeneity of existing repositories. Therefore, this evaluation would lead to more reliable and reproducible performance results of a system. However, most of the publicly available datasets contain ADLs and falls induced in laboratory controlled conditions rather than in free-living conditions [57]. In this regard, repositories such as FARSEEING contain real-world fall data. Nevertheless, as that dataset is private, the use of the full dataset information is limited to researchers who collaborate with the FARSEEING repository [58]. However, it is important to mention that advances have been performed during the latest years in order to decrease the gap between laboratory-induced and real-world falls [59].

According to the search results, the validation process is mostly achieved either by using CV [8,9,26,27,28,29] or Holdout [5,25,30] methods. Despite its simplicity, the Holdout method produces a reduced dataset for algorithms’ training and testing, which could lead to a generation of weaker models and a smaller dataset to test its classification performance [18]. CV emerged as an alternative, as it would substantially increase the data available for algorithm training and testing. This validation method became widely used to estimate the generalisation performance of Machine Learning models [18,60]. This is in line with the search results obtained, considering that more than half of the validation methods applied were related to CV [8,9,26,27,28,29]. It is noteworthy that none of the studies used the resubstitution method to validate the fall risk assessment systems performance. In this methodology, the model is trained and tested with the same dataset, leading to an obvious overfitting of the model towards the validation dataset and over-optimistic performance results [18]. In fact, Shany et al. [18] identified some studies that performed this inefficient validation model. Thus, it is possible to understand that recent work on fall risk assessment systems has been addressing more robust validation methods, disregarding weaker methods such as resubstitution.

Overall, the performance results obtained by fall risk assessment systems were quite promising. Regarding fall risk assessment based on clinical scales, various studies reported high performance from their systems. The Deep Learning model developed by Rivolta et al. [25] achieved a sensitivity and a specificity of 86% and 90%, respectively, towards the classification of individuals at high or low fall risk category based on the Tinetti test score attributed at the baseline assessment. In addition, Saporito et al. [27] and Shahzad et al. [26] obtained a relatively low misclassification error towards the estimation of participants’ TUG and BBS clinical scale scores, respectively. The smart insole system developed by Yang et al. [31] also showed high values of accuracy in estimating relevant spatio-temporal features from the TUG test that enable the assessment of fall risk. Concerning fall risk assessment based on the detection of fall risk events, Saadeh et al. [4] obtained an outstanding performance detecting fall risk events, reporting a sensitivity of 97.8% and a specificity of 99.1%. Leone et al. [5,29] also achieved accuracy, sensitivity, and specificity values between 80% and 90%. Nevertheless, as previously mentioned by Shany et al. [18], fall risk assessment study results are often over-optimistic considering the reduced number and age of subjects enrolled in the test. In addition, even the pervasively used CV presents some problems given the fact that its statistical properties are not fully understood [18,60]. Furthermore, a remarkable lack of external validation of fall risk assessment systems was observed. These topics should be further addressed and discussed in future studies in order to reliably design and validate fall risk assessment systems while tackling the limitations and gaps found in current studies.

### 4.6. Future Directions and Work

As the interest in the field of fall risk assessment is growing, it is expected that novel wearable monitoring solutions will emerge and enhance the assessment’s performance. That can be enabled by: (i) the advances on the current used sensing technologies; (ii) the used algorithms; or (iii) the introduction of innovative wearable sensors that record meaningful data for this assessment. Regarding this last topic, the advances of sensors that measure biosignals can play an important role by providing meaningful metrics underlying a subject’s biomechanical reactions to falls. Future work on the fall risk assessment field may focus on a multifactorial approach to assess the risk of fall, comprising meaningful data provided by wearable kinematic, kinetic, and biosignal sensors [2]. Nevertheless, it is essential to perform a trade-off between the number of sensors used, which should be the lowest number possible, and the system’s algorithm performance, that should be as high as possible. Fall risk assessment systems must be user-centred designed so that the user feels compliant with the designed sensor system, in order to be able to use it for long periods of time without any issues [2].

According to the topics previously discussed, a solution to accomplish a comprehensive fall risk assessment may be a system that: (i) monitors the risk of fall in real-time, based on the detection of fall risk events; and (ii) has the option to predict the score of the most suited clinical established scales, in order to conduct a long-term prediction of the individual’s fall risk. This long-term evaluation may motivate the subject to decrease its fall risk by being able to compare its current clinical scale index of fall risk with the previous ones obtained. The ideal scenario is that all of this assessment is executed during the everyday life and that the user does not need to go to any medical care centre to perform clinical scales towards fall risk assessment. However, despite the encouraging performance of the real-time fall risk assessment systems towards the timely detection of fall risk events, its applicability to accurately prevent falls in the elderly community remains unclear. The elderly may not be agile enough to react to a fall risk alarm and prevent a fall, considering their level of physical and cognitive decline and how rapidly a fall occurs [1]. In fact, to the authors’ knowledge, there are no studies in the scientific literature that address and evaluate the applicability of fall risk assessment systems to actually prevent falls. In this regard, two potential solutions could be used with the fall risk assessment systems in order to enhance the likelihood of balance recovery upon a fall risk event: (i) trigger an assistive system attached to the subject, whenever a fall risk event is detected, in order to help regain balance and thus prevent the fall [61,62]; or (ii) improve subject’s reactive stability and fall resisting skills. This can be achieved through conventional training, such as Tai-chi, which has proven effective towards fall prevention by improving balance, muscle strength, endurance, and proprioception [63]. Nevertheless, perturbation-based balance training (PBT), which is a promising new task-specific training, has also been shown to reduce fall incidence [64]. Essentially, PBT consists on the delivery of unexpected destabilising balance perturbations during walking, which match real-life loss of balance scenarios, in a controlled environment [59,64,65]. The goal of this training scheme is to prepare high fall risk subjects to develop fall resisting skills to counteract real-life loss of balance events. When using an assistive device as a means to prevent a fall, several considerations have to be researched to verify their applicability. Falls happen very fast. Thus, the applicability of a system to prevent a fall must be assessed to guarantee that after the detection of the incoming signal to prevent a fall, there is still enough remaining time to prevent it.

It is also necessary to plan and perform a suitable and reliable validation of the performance of the fall risk assessment systems [18]. Hence, future work should also focus on the identification of gold standard external validation sources, i.e., public datasets, in which systems could be benchmarked. This would provide a reliable comparison between the different literature fall risk assessment systems. In this regard, as these systems are intended to be used by the elderly or subjects with mobility deficits, an effort should be performed to validate the systems with data collected from these target populations.

## 5. Conclusions

The current *state-of-art* of fall risk assessment systems analysed in this narrative review showed that most of the studies performed fall risk assessment based on clinical scales. In the studies within this group, kinematic and kinetic data collected by inertial and pressure sensors, respectively, were the most widely used sensing modalities, and sensors were generally placed in the upper body. In the studies that performed fall risk assessment based on the detection of fall risk events, it was identified a trend to use EMG sensors on lower limb muscles. Both identified methods seem to preferably adopt Machine Learning models to classify the subject’s risk of fall. Concerning fall risk assessment systems validation, it was shown that the number of participants enrolled in the studies’ experimental protocols was reduced. In addition, some studies did not include elderly participants. CV was found to be the most adopted validation method. The lack of external validation was remarkably noticed, considering that almost all studies performed internal validation of the developed systems. Validation results suggested that an acceptable performance was obtained by some fall risk assessment systems. However, we identified the need for the establishment of an open access gold standard by which different fall risk assessment systems could be benchmarked. This would pave the way for a reliable performance comparison between the different systems developed in the literature.

## Figures and Tables

**Figure 1 sensors-22-00984-f001:**
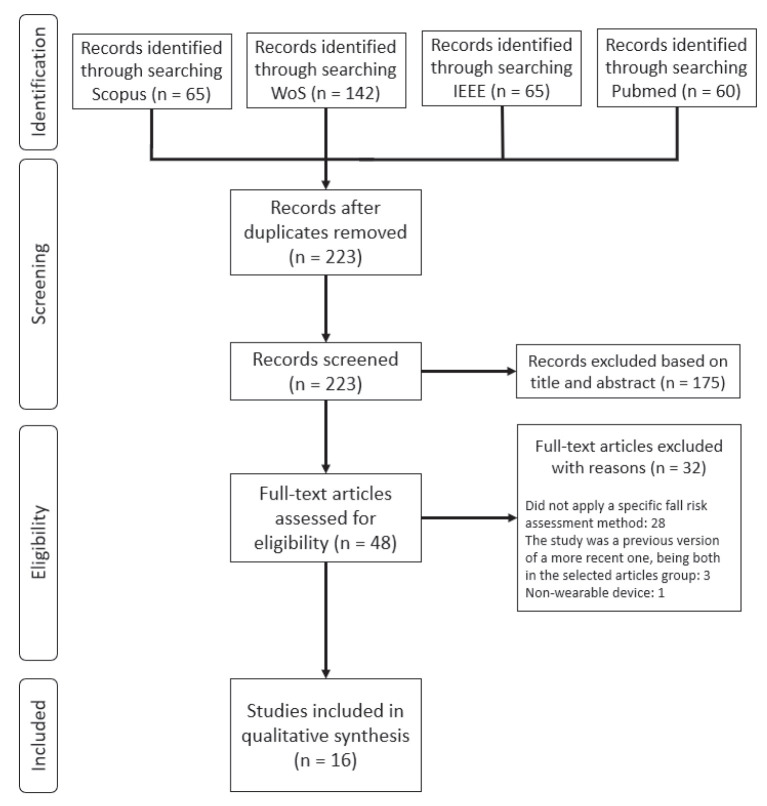
PRISMA flow diagram.

**Figure 2 sensors-22-00984-f002:**
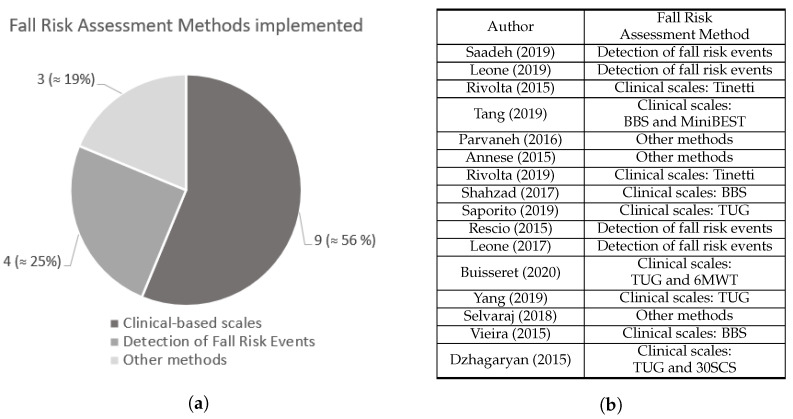
(**a**) Number of studies from each fall risk assessment methods identified. (**b**) Fall risk assessment method adopted by each study. Saadeh [4], Leone [5], Rivolta [8], Tang [9], Parvaneh [12], Annese [13], Rivolta [25], Shahzad [26], Saporito [27], Rescio [28], Leone [29], Buisseret [30], Yang [31], Selvaraj [32], Vieira [33], and Dzhagaryan [34].

**Figure 3 sensors-22-00984-f003:**
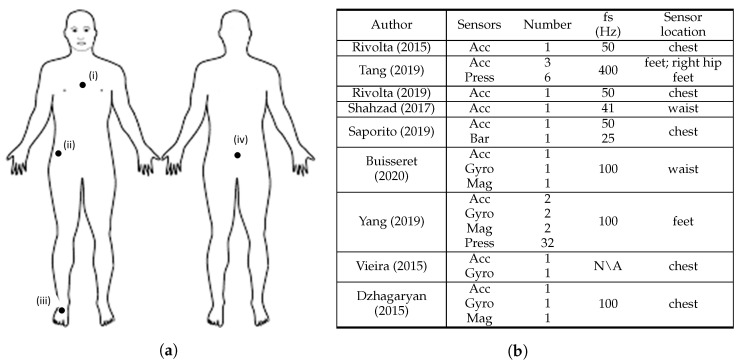
Overview of the sensor characteristics from clinical scale-based fall risk assessment studies. (**a**) Anterior and posterior views of the human body depicting sensor location, where: (i) [8,25,27,33,34], (ii) [9], (iii) [9,31], and (iv) [26,30]. (**b**) Adopted sensor specifications, where: S = sensors, N = number, fs = sampling frequency, Acc = accelerometer, Gyro = gyroscope, Mag = magnetometer, Press = pressure sensors, Bar = barometer, Dist = distance sensors, N\A = Not Available. Rivolta [8], Tang [9], Rivolta [25], Shahzad [26], Saporito [27], Buisseret [30], Yang [31], Vieira [33], and Dzhagaryan [34].

**Table 1 sensors-22-00984-t001:** Sensor characteristics from the fall risk assessment studies based on the detection of fall risk events, where: fs = sampling frequency, Acc = accelerometer.

Author	Sensors	Number	fs (Hz)	Sensor Location	Mean Lead-Time (ms)	Lead-Time Meaning
Saadeh [4]	Acc	1	256	upper thigh	300–700	time between the detection of the unbalance event and the impact of the fall
Leone [5]	EMG	4	125	gastrocnemius and tibialis muscles	750	time between the detection of the unbalance event and the impact of the fall
Rescio [28]	EMG	4	1000	gastrocnemius and tibialis muscles	200	time difference between the perturbation onset and the detection of the perturbation
Leone [29]	EMG	4	1000	gastrocnemius and tibialis muscles	775	time between the detection of the unbalance event and the impact of the fall

**Table 2 sensors-22-00984-t002:** Validation characteristics adopted by the 11 selected articles, where: ML = machine learning, Th = threshold-based, Accu = accuracy, Sens = sensitivity, Spec = specificity, CV = cross-validation, NLSVM = NonLinear Support Vector Machine classifier, LDA = Linear Discriminant Analysis classifier, SVR = Support Vector Regression, ANN = Artificial Neural Networks, LLS = Linear Least Square Regression, LASSO = Least Absolute Shrinkage and Selection Operator regression, and CNN = Convolutional Neural Network.

Author	Number of Subjects	Subject’s Age	Model Used	Validation Method	Reference Measures for Classification	Results
Saadeh [4]	77	20-70	ML (NLSVM)	N\A	Type of event (pre-fall or normal ADL events).	Sens = 97.8%; Spec = 99.1%
Leone [5]	5	28.7 ± 7.1	ML (LDA)	Holdout (70% training; 30% testing)	Type of event (pre-fall or normal ADL events).	Accu = 82.3%; Sens = 86.4 %; Spec = 83.8%
Rivolta [8]	13	69.7 ± 10.7	ML (multiple linear regression model)	Leave-one-out CV	Clinical score (Tinetti)	Accu = 84.6% Sens = 85.7%; Spec = 83.3%
Tang [9]	30	76.0 ± 10.5	ML (Linear kernel SVR)	Leave-one-out CV	Clinical score (BBS and MiniBEST)	Mean error: 6.07 ± 3.76 (BBS); 5.45 ± 3.65 (MiniBEST)
Rivolta [25]	90	69.3 ± 16.8	ML (linear regression model); DL (single hidden layer ANN)	Holdout (60% training; 40% testing)	Clinical score (Tinetti)	Sens (ML) = 71% Spec (ML) = 81% Sens (DL) = 86%; Spec (DL) = 90%
Shahzad [26]	23	72.87 ± 8	ML (LLS and LASSO models)	10-fold CV	Clinical score (BBS)	Mean error: 1.9 ± 2.53 (LLS); 1.44 ± 1.98 (LASSO)
Saporito [27]	239	75.2 ± 6.1	ML (regularised linear model)	Leave-one-out CV	Clinical score (TUG)	Mean error: 2.1 ± 1.7s
Rescio [28]	7	28.8 ± 7.6	Th	10-fold CV	Type of event (pre-fall or normal ADL events)	Sens 70%; Spec 70%
Leone [29]	15	32.6 ± 9.3	ML (LDA)	10-fold CV	Type of event (pre-fall or normal ADL events)	Sens = 89.1%; Spec = 87.1%
Buisseret [30]	73	83.0 ± 8.3	Th; DL (CNN)	Holdout (78% training; 22% testing)	Faller status based on prospective fall occurrence	Accu(Th) = 73.9%; Sens(Th) = 85.7%; Spec(Th)= 50%; Accu(DL) = 75%; Sens(DL) = 75%; Spec(DL) = 75%
Yang [31] (*)	10	19-44	N\A	N\A	Video recordings from TUG	Accu(gait cycle count) = 100% Accu(segment TUG phases) = 92.23% Accu(spatial—temporal features) = 92%

(*) This study validated a system that extracted features from TUG rather than directly validate the system towards the classification of fall risk.

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
