# Peer review of "Fall Risk Assessment Using Wearable Sensors: A Narrative Review"

_sensors, 2022, doi:10.3390/s22030984_

Round 1

Reviewer 1 Report

Dear authors,

thank you very much for proposing your narrative review "Fall Risk Assessments using Wearable Sensors: A narrative review" to Sensors.

Overall, this is a very well written article which I think is a great contribution to this topic. I have some minor comments which I think are important to consider:

1) Page 2, line 84: You described older people as walking in a "safer" way when having experienced a prior fall. This is misleading in my opinion. While it is known that older people tend to change their activity any walking patterns after experiencing a fall, the term "safe" is wrong. Rather, observable changes in gait patterns and muscle activity pattterns (heightened levels of co-contraction, reduced step length and cadence, cahnges in One-Leg and Two-Leg standing times, among others) due to fear of falling are indeed raising the risk of falling long-term. This is important to understand because, especially in the field of sensor based gait measurement in a free-living environment, such changes are important to detect in interpret appropriately. I therefore suggest to takke these factors into account in your introduction.

2) Page 3, line 136: Is there reason why you narrrowed your search strategy to articles published after 2015? Please include your reasoning in your description of your search strategy.

3) Page 7, Line 276-277: Here, you mention the predictability of the sensor systems to detect a fall. At this point, I was wondering about any details about the accuracy, such as Sensitivity / Specifity. These details are presented in table 2, so I would suggest to refer to this table at this point.

4) One general observation to your discussion an fall prediction systems: I think you clearly described the abiliy of these sensor systems to accurately detect a fall. While this ability of these systems are important as a prerequisite to any application in a clinical or free-living setting, my main concern is that the applicability of such systems to prevent a fall are, to put it mildly, unclear. Falls happen very fast. If a sensor system is able to detect the occurance of a fall between 200 and 700 ms before the actual fall, I have some serious doubts that an older person with a level of  physicall decline that is typically accompanied with a risk of falling, is able to react fast enough to any incoming signal to prevent a fall at this point. To my knowledge, studies on the applicability of such sensors in a sense that the ability of these sensors to prevent falls is evaluated, are lacking. Because this is a really important point for clinicians (like me), I would like you to expand your discussion on this matter.

I hope that these points help you in your further process of publishing your narrative review.

Reviewer 2 Report

Recommendation Accept in present form.

The manuscript is ready for the press. 

One word comes to this reviewer's mind and that is "Value".

This publication should be kept by one's side as a reference to use to assess and choose rational options among the list of fall-risk wearable clinical sensors.

Congratulations!

Reviewer 3 Report

In this manuscript, a review for the fall risk assessment using wearable sensors has been presented, where different fall risk assessment methods performed in scientific literature using wearable sensors are identified and evaluated. The review is interesting and important. The manuscript is well organised and written. However, the future directions and work part need to be enhanced.

Round 2

Reviewer 3 Report

I am satisfied with the revision and have no further comments.